# Ammonia Volatilization and Marandu Grass Production in Response to Enhanced-Efficiency Nitrogen Fertilizers

**Juliana Bonfim Cassimiro** [1,*], **Clayton Luís Baravelli de Oliveira** [2], **Ariele da Silva Boni** [1], **Natália de Lima Donato** [1], **Guilherme Constantino Meirelles** [1], **Juliana Françoso da Silva** [1], **Igor Virgilio Ribeiro** [1] **and Reges Heinrichs** [1]

1   Department of Crop Science, São Paulo State University—UNESP, Rodovia SP 294, km 651, Dracena 17900-000, SP, Brazil
2   Soil and Fertilizers Laboratory, Universidade do Oeste Paulista—UNOESTE, Rodovia Raposo Tavares, Presidente Prudente 19067-175, SP, Brazil
*   Correspondence: bonfimjuliana70@gmail.com

**Abstract:** The objective of this study was to evaluate dry matter (DM) production of *Urochloa brizantha* cv. Marandu and ammonia volatilization in response to rates and sources of enhanced-efficiency N fertilizers. The experiment was took place in a pasture area, two growing seasons. A randomized block design with four replications was used, in a $4 \times 2 + 1$ factorial arrangement, consisting of four N sources (Urea—$Ur_{Conv}$; Ammonium nitrate—AN; Urea + NBPT—$Ur_{NBPT}$; Urea + Duromide—$Ur_{Duromide}$) and two nitrogen rates (100 and 200 kg ha$^{-1}$ year), plus a treatment without nitrogen fertilization (control). At both N rates, ammonia volatilization from $Ur_{Conv100/200}$ was greatest. Ammonia volatilization was less after $Ur_{NBPT}$ and $Ur_{Duromide}$ application, with values similar to AN. Ammonia losses from $Ur_{Duromide}$ tend to be lower than from $Ur_{NBPT}$. The N use efficiency in dry matter production of Marandu was influenced by the N sources and rates. At both N rates, the efficiency of $Ur_{Duromide}$ and $Ur_{NBPT}$ was greater than that of $Ur_{Conv}$. With regard to total DM and leaf percentage in response to N rates, DM production increased after 200 kg N ha$^{-1}$ rates in response to all sources, in both years. The $Ur_{Duromide}$ reduce N losses by volatilization compared to $Ur_{NBPT}$ and $Ur_{conv}$, and resulted in greater total DM production and relative leaf production of Marandu, in comparison to $Ur_{NBPT}$, AN and $Ur_{conv}$.

**Keywords:** dry matter production; Duromide; pasture; urease inhibitor; *Urochloa brizantha*

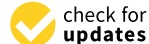



## 1. Introduction

*Urochloa brizantha* cv. Marandu is native to tropical Africa and adapts well to soils of medium fertility, obtaining high yields in fertile soils [1]. With a short cycle and perennial, it grows in clump form, its stems have a dense hairiness, and the plant has good digestibility and palatability. When grown in medium to high fertility soils, the plants exceed 1.5 m in height [2]. The predominance of the genus Urochloa in Brazil is more common in sown pastures, with about 50 million hectares of arable land, mainly because it is robust, and is associated with a high productive potential, great nutritional quality and widely adaptable in various edaphoclimatic environments [3–5].

The forages used for pasture, extremely relevant for livestock production, are predominantly grasses, which are very responsive to nitrogen fertilization. This nutrient (N) is applied in large amounts to pasture, so sound agronomic practices and modern technology are needed in the sector [6].

Nitrogen fertilization can improve yield and crude protein content of forages [7,8]. Urea is the most frequently used nitrogen source, mainly due to its low cost. However, it is highly susceptible to losses by ammonia volatilization, caused by changes in soil moisture, temperature and pH, wind speed, soil organic C and N, and the applied urea rate [9].

Certain products have increase urea efficiency, e.g., urea treated with N–(n-butyl) thiophosphoric triamide (NBPT), marketed since 1996 in the United States and, more recently, in Brazil. This urease inhibitor is currently the only commercially available option for agriculture and is sold in more than 70 countries [10,11]. The main benefit of stabilizing urea by adding substances for this purpose is the reduction of volatilization, which: (a) extends the period (number of days) of chemical stability between nitrogen fertilization and soil incorporation by rainwater or irrigation, reducing N losses by volatilization; (b) reduces the N volatilization losses caused by urea hydrolysis on the soil surface; (c) increases N uptake, fertilization efficiency and crop yield and quality [10].

The retardant NBPT is a conventional inhibitor that loses its efficiency under acidic soil pH conditions and temperatures above 30 °C. A novel urease inhibitor called Duromide is the active ingredient of a new urease inhibitor generation [12]. The new molecule has the same chemical function as conventional NBPT, having the same mode of action to block the urease active site by binding to it. On the other hand, the rest of its chemical structure differs from NBPT, making it more stable. The greater stability of this new active principle raises expectations of allowing more durable storage and applications in wider ranges of soil pH and temperature. Based on these characteristics, enhanced urease inhibition and accordingly, reduced N volatilization losses are expected [12]. In view of the above, the objective was to evaluate ammonia volatilization and dry matter production of *Urochloa brizantha* cv. Marandu in response to rates and sources of enhanced-efficiency N fertilizers.

## 2. Material and Methods

### 2.1. Experimental Area

The experiment was installed in an experimental field of the Faculty of Agricultural and Technological Sciences of UNESP, Campus de Dracena (21°27′ S; 51°36′ W), with a tropical climate, classified as Aw by Köppen [13], mean annual rainfall of approximately 1300 mm, mean annual air temperature of 24 °C and mean maximum of 31 °C and minimum of 19 °C. An area in the process of pasture formation of *Urochloa brizantha* cv. Marandu was evaluated over two growing seasons (2018/2019 and 2019/2020) (Figure 1).

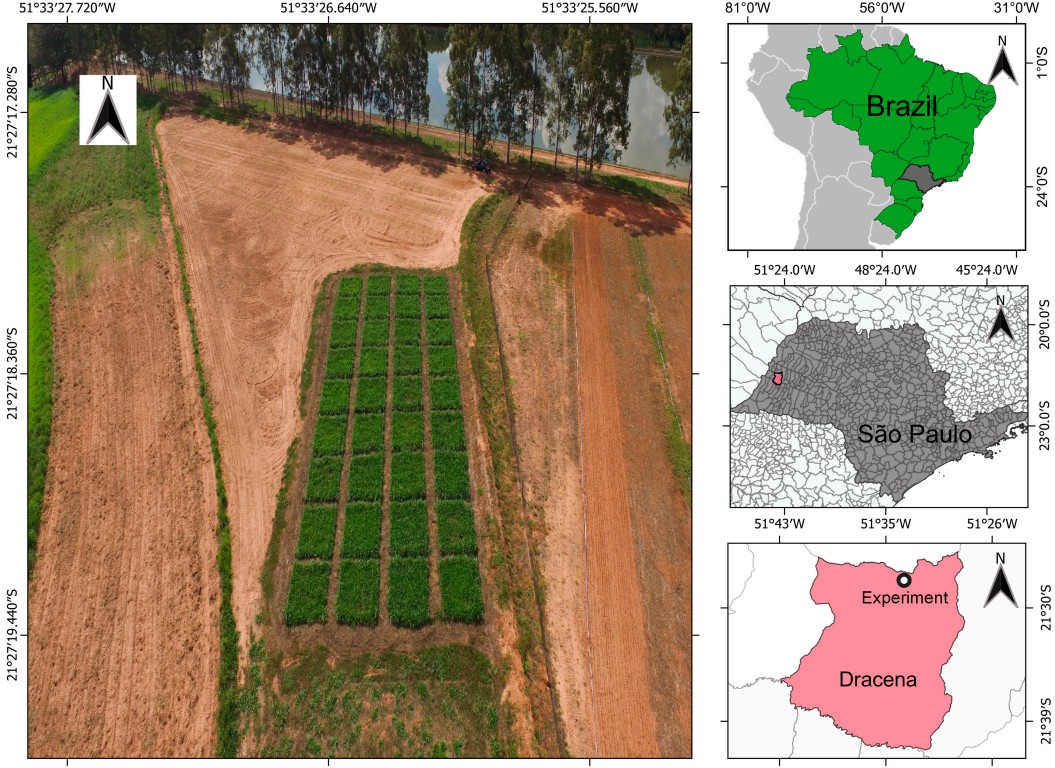

**Figure 1.** Location of experimental area, Dracena-SP, Brazil.

The soil of the experimental area was classified as Argissolo Vermelho Amarelo distrófico soil with sandy texture [14], corresponding to a dystrophic Ultisol [15]. Soil chemical and particle-size analyses (0.00–0.20 m layer), showed the following results: 13 g dm$^{-3}$ organic matter; pH (CaCl$_2$) 4.5; 3 mg dm$^{-3}$ P (Resin); 5.0 mmol$_c$ dm$^{-3}$ Ca$^{2+}$; 3.0 mmol$_c$ dm$^{-3}$ Mg$^{2+}$; 1.4 mmol$_c$ dm$^{-3}$ K$^+$; 24.4 mmol$_c$ dm$^{-3}$ cation exchange capacity; 120 g kg$^{-1}$ clay; 30 g kg$^{-1}$ silt; and 850 g kg$^{-1}$ sand.

### 2.2. Experimental Design

The experiment was arranged in randomized blocks with four replicates, in a $4 \times 2 + 1$ factorial design, consisting of four N sources: conventional urea (Ur$_{Conv100}$ and Ur$_{Conv200}$), ammonium nitrate (AN$_{100}$ and AN$_{200}$), urea treated with NBPT (Ur$_{NBPT100}$ and Ur$_{NBPT200}$), urea treated with Duromide (Ur$_{Duromide100}$ and Ur$_{Duromide200}$) and two nitrogen rates (100 and 200 kg ha$^{-1}$ year), plus a treatment without nitrogen fertilization (control), resulting in a total of 36 plots. The annual rates of 100 and 200 kg N ha$^{-1}$ were split in four applications of 25 and 50 kg ha$^{-1}$, respectively, broadcast on the soil surface. The first fertilization was applied 30 days after sowing and the others subsequently after each cut. In the second year, the first rate was applied in the beginning of the rainy season, in October 2019, and the others after the next three cuts, resulting in a total of four applications.

### 2.3. Soil Management, Sowing and Cultural Treatments

In August 2018, dolomitic limestone was incorporated to a soil depth of 0.20 m, to raise base saturation to 60%. Three months after liming, immediately before sowing, 80 kg P$_2$O$_5$ ha$^{-1}$ as single superphosphate and 30 kg K$_2$O ha$^{-1}$ as potassium chloride were broadcast [16].

Marandu grass was sown in rows (December 2018), spaced 25 cm apart, at a density of 10 kg ha$^{-1}$ of pure, healthy seeds. The plot size was $4 \times 4$ m and spacing between plots and blocks was 1 m.

In January 2019, the first nitrogen fertilization was carried out in the treatments, at the respective rates. In each growing season, four cuts were made in the rainy and one in the dry season, resulting in a total of five forage cuts per growing season. In the second year, the same phosphorus and potassium rates as in the first were applied, together with the first N fertilizer application.

### 2.4. Ammonia Volatilization

Volatilization cylinders similar to those described by [17–20] were used (Figure 2). In the first year of evaluation, ammonia volatilization was evaluated by analyzing the polyethylene foam strips soaked in phosphoric acid, which were collected from the cylinders and exchanged on day 2, 5, 9, 14, 20 and 26 after fertilization with each of the four N rates. In the second year, these foams were collected and exchanged twice (tow cycles), on day 1, 2, 5, 9, 14, 20 and 26 after the 2nd and 4th application of N rates, to assess ammonia volatilization.

The polyethylene cylinders were fitted on top of round PVC bases (diameter 9 cm, height 10 cm) (Figure 2). In each plot, six bases per cylinder were installed (one per evaluation). Since the contact between rain and fertilizer was impeded within the cylinders, they were shifted to a subsequent base at each foam exchange. In this way, in the following period, NH$_3$ losses from the fertilizer treatments were evaluated under exposure to the same conditions (rain, temperature, wind, etc.) as in the rest of the experimental field.

At the moment of fertilizer broadcasting in the total area of the plots, the volatilization cylinders were sealed and the respective relative amount of fertilizer of each treatment was individually weighed and applied to the area within the bases underneath the cylinders. In each cylinder, the amount of fertilizers applied corresponded to rates of 100 and 200 kg N ha$^{-1}$, divided into four applications of 25 and 50 kg ha$^{-1}$, respectively, per growing cycle.

To determine volatilization, the retained ammonia was extracted from the foam strips by washing four times with 10 mL deionized water and measuring this solution in a 100 mL volumetric flask, Thereafter, an aliquot of 20 mL was distilled and the volatilized $NH_3$ was determined by subsequent titration ($H_2SO_4$ 0.0025 mol $L^{-1}$) [21].

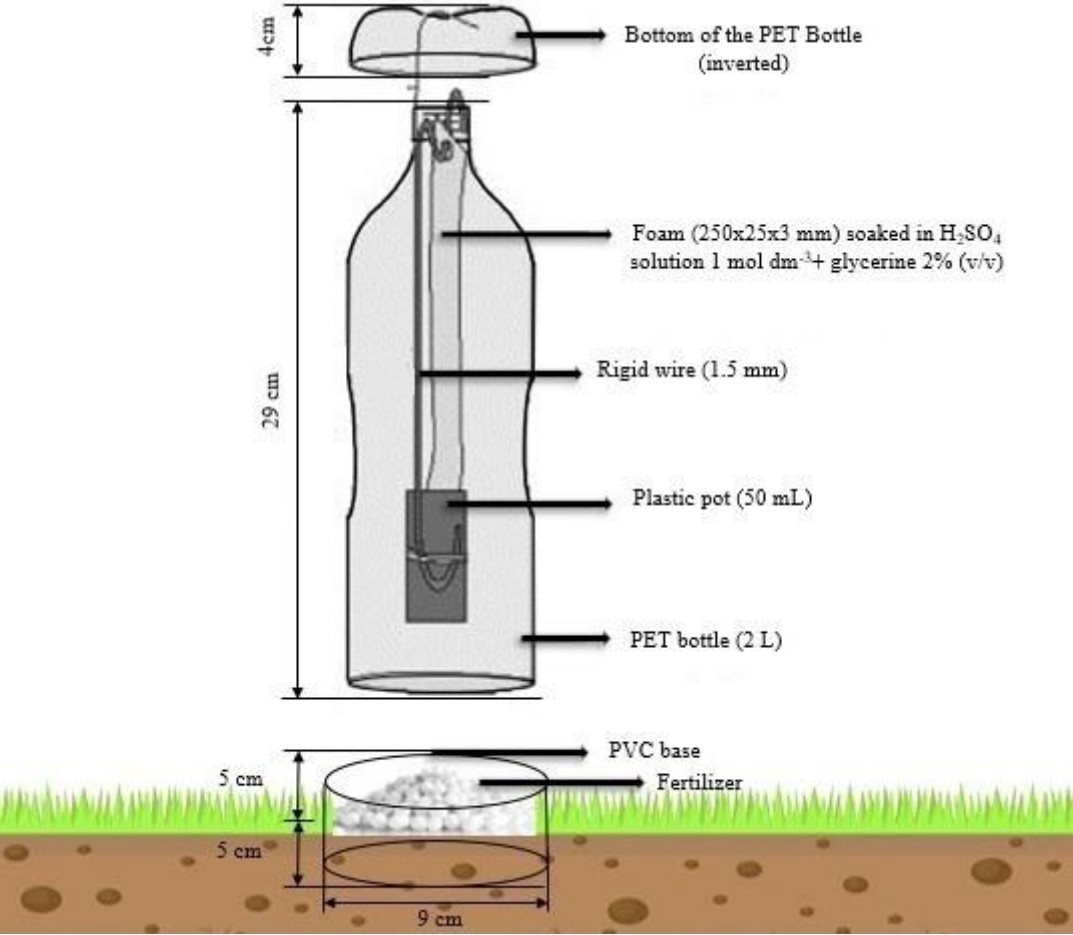

**Figure 2.** Semi-open static $NH_3$ collection cylinder.

### 2.5. Grass Production

Four cuts were taken, at intervals regulated by the mean plant height of 28 cm (i.e., 95% light interception) in the best treatment [22]. In the rainy season, the cutting height was reached within intervals of approximately 30 days.

Leaf fresh matter was measured in one sample of 0.5 $m^2$ (1 × 0.5 m) per experimental unit, taken with a rectangular iron sampler. The sampler was placed randomly at representative points of each plot and the forage within the rectangle was cut at 15 cm above the ground and immediately weighed to determine leaf fresh matter. Then, a forage subsample was taken, immediately weighed and dried to constant weight in a forced air circulation oven at 65 °C, to determine dry matter production [23].

### 2.6. Morphological Composition of Forage

To determine the relative participation of each morphological component, a subsample of leaf fresh matter was removed, separated in leaf blades and pseudostem (stems + sheaths) and separately dried to constant weight in a forced-air circulation oven, at 65 °C [23].

### 2.7. Data Analysis

Data were evaluated for error normality and homogeneity and the results subjected to analysis of variance and mean comparison by the Tukey test at 5% significance ($p < 0.05$).

Pearson's correlation (Minitab statistical software version 21.2.0, State College, PA, USA) was calculated to investigate the relationship between total dry matter, leaf percentage, leaf dry matter, N loss reduction, N use efficiency and percent N loss. All statistical analyses were performed using the Statistical Analysis System software (SAS OnDemand for Academics 2022). Graphs were plotted using Sigmaplot® version 14.5 (Systat Software, Inc., San Jose, CA, USA, www.sigmaplot.com, accessed on 12 December 2022). The model was selected according to the Akaike Information Criterion (CIA) [24] by choosing the models with the least CIA. After selecting the model, the data were subjected to non-linear regression, using the logistic model represented by Equation (1), as described by [25]. This model is traditionally used to estimate cumulative ammonia volatilization [26–28].

$$\hat{Y} = \frac{\alpha}{1 + exp^{[-(t-\beta)/\gamma]}} \tag{1}$$

where $\hat{Y}$ is the amount of N volatilized in form of $NH_3$-N (kg ha$^{-1}$) at time t; $\alpha$ is the maximum cumulative volatilization; $\beta$ the moment when 50% of the losses had occurred, corresponding to the inflection point of the curve (day of maximum daily $NH_3$-N loss); $t$ the time (days); and $\gamma$ a parameter of the equation used to calculate the maximum daily loss (MDL) of $NH_3$-N, as shown in Equation (2).

$$MDL = \frac{\alpha}{4\gamma} \tag{2}$$

To evaluate the reduction in ammonia loss compared to urea, as shown in Equation (3).

$$NH_3 - N = 100 - \frac{N\ loss\ by\ fertilizer \times 100}{N\ loss\ by\ urea} \tag{3}$$

## 3. Results

### 3.1. $NH_3$-N Volatilization Losses

Figure 3a,c,e,g and Figure 4a,c show ammonia volatilization over a 26-day period. The climatic conditions of each evaluation period are shown in Figure 3b,d,f,h and Figure 4b,d. In the four evaluations of both annual N rates, ammonia volatilization from $Ur_{Conv}$ was observed to be highest. In turn, $Ur_{NBPT}$ and $Ur_{Duromide}$ proved more efficient in reducing ammonia volatilization compared to conventional urea, reaching values close to AN.

In the first experiment (2019) (Figure 3a), two days after the first split applications of $Ur_{Conv100}$ and $Ur_{Conv200}$, respectively, 2.82 and 7.45 kg ha$^{-1}$ of the applied N was lost, with losses peaking on the 5th day after fertilization, with 5.41 and 13.93 kg N ha$^{-1}$, respectively. Subsequently, the N losses from $Ur_{NBPT200}$ reached approximately 6.52 kg N ha$^{-1}$, 3.76 kg N ha$^{-1}$ from $Ur_{Duromide200}$ and approximately 0.32 kg N ha$^{-1}$ from $AN_{200}$. In the first experiment, $NH_3$ losses were less after application of fractional rates of 25 kg N ha$^{-1}$ (Table 1). Compared with urea, $NH_3$ losses from AN, $Ur_{NBPT}$ and $Ur_{Duromide}$ for the split rates of 25 kg N ha$^{-1}$ were reduced by 94.4%, 35.1% and 52.4%, respectively, and by 97.7%, 54.2% and 74.7% in response to rates of 50 kg N ha$^{-1}$ (Table 1).

The second evaluation of the first year (Figure 3c) showed that 3.94 and 7.83 kg N ha$^{-1}$ was lost from $Ur_{Conv100/200}$ on the second day of data collection, while the sources $Ur_{NBPT100/200}$ and $Ur_{Duromide100/200}$ lost approximately 1 kg N ha$^{-1}$. During this period, it should be mentioned that rain (8 mm) fell on the first day after fertilization and another rainfall (7 mm) occurred on the second day, after data collection (Figure 3d). The reduction in ammonia loss in this cycle was greater than in the first, reaching 96.2% and 98% at rates of 25 and 50 kg N ha$^{-1}$ compared to AN. This performance was better than that of $Ur_{NBPT100/200}$, with respective reductions of about 73% and 81.4% and for $Ur_{Duromide100/200}$, with respective reductions of about 81.9% and 87%, exceeding those in response to $Ur_{NBPT}$ (Table 1).

After the third fertilization (Figure 3e,f), the first rainfall occurred only nine days after application, and during this rain-free period, volatilization was practically nonexistent. As

of the onset of rainfall (37.6 mm), intense volatilization occurred in response to $Ur_{Conv100/200}$, peaking at 14 days after fertilization, with respective losses of 4.04 and 10.43 kg N ha$^{-1}$ at the end of the cycle. Losses were least from $Ur_{Duromide100}$ (0.42 kg N ha$^{-1}$), and $Ur_{Duromide100}$ reached a 90.8% reduction in NH$_3$ losses (Table 1). The efficiency of urease inhibitor with urea in reducing ammonia volatilization was confirmed. Thus, N volatilization losses from unprotected urea proved to be higher than from urea with urease inhibitor, in other words, volatilization losses from $UrDuroimide_{100/200}$ were 90.8% and 88.4% less than from $Ur_{Conv100/200}$ (Table 1).

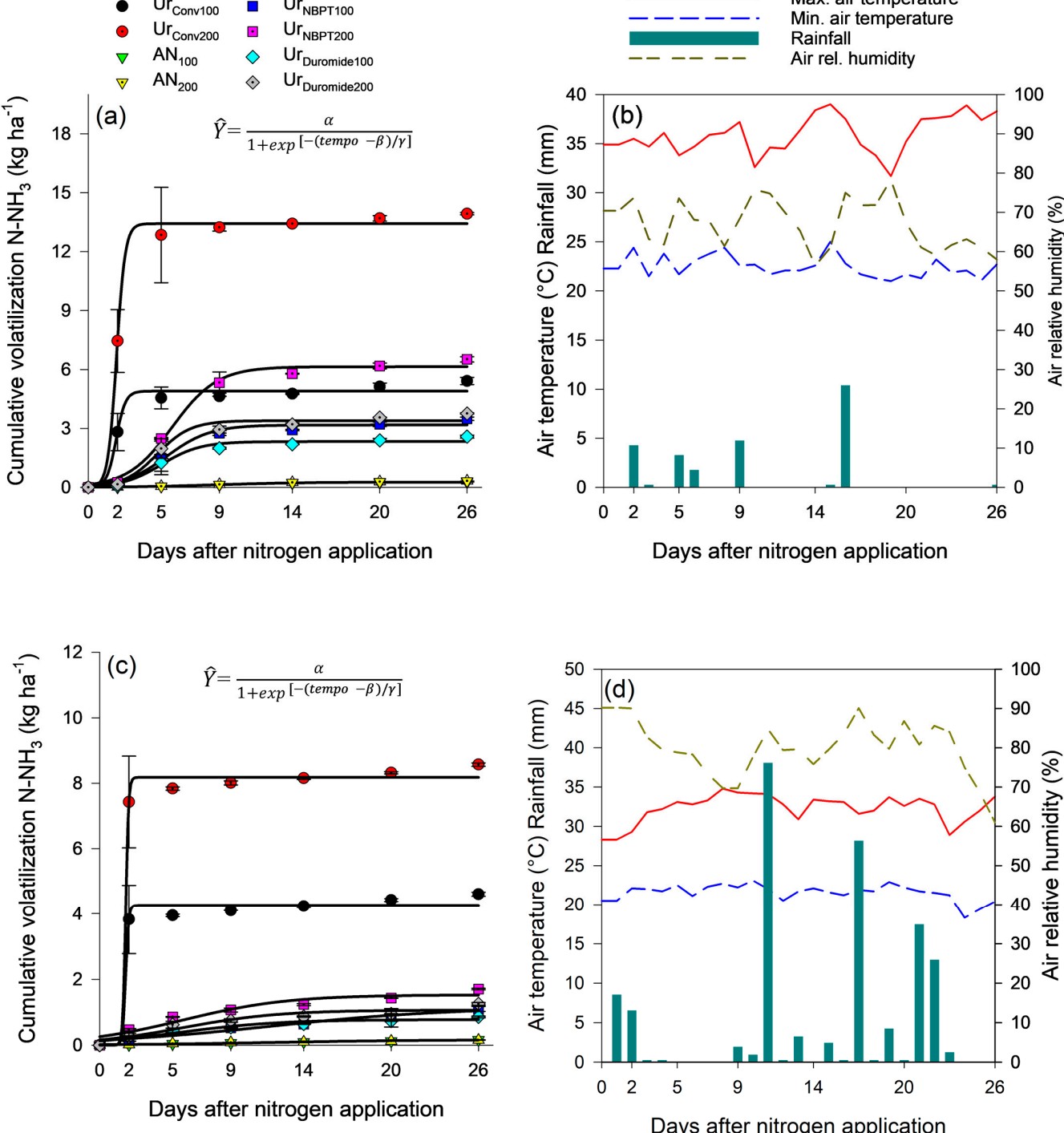

**Figure 3.** *Cont.*

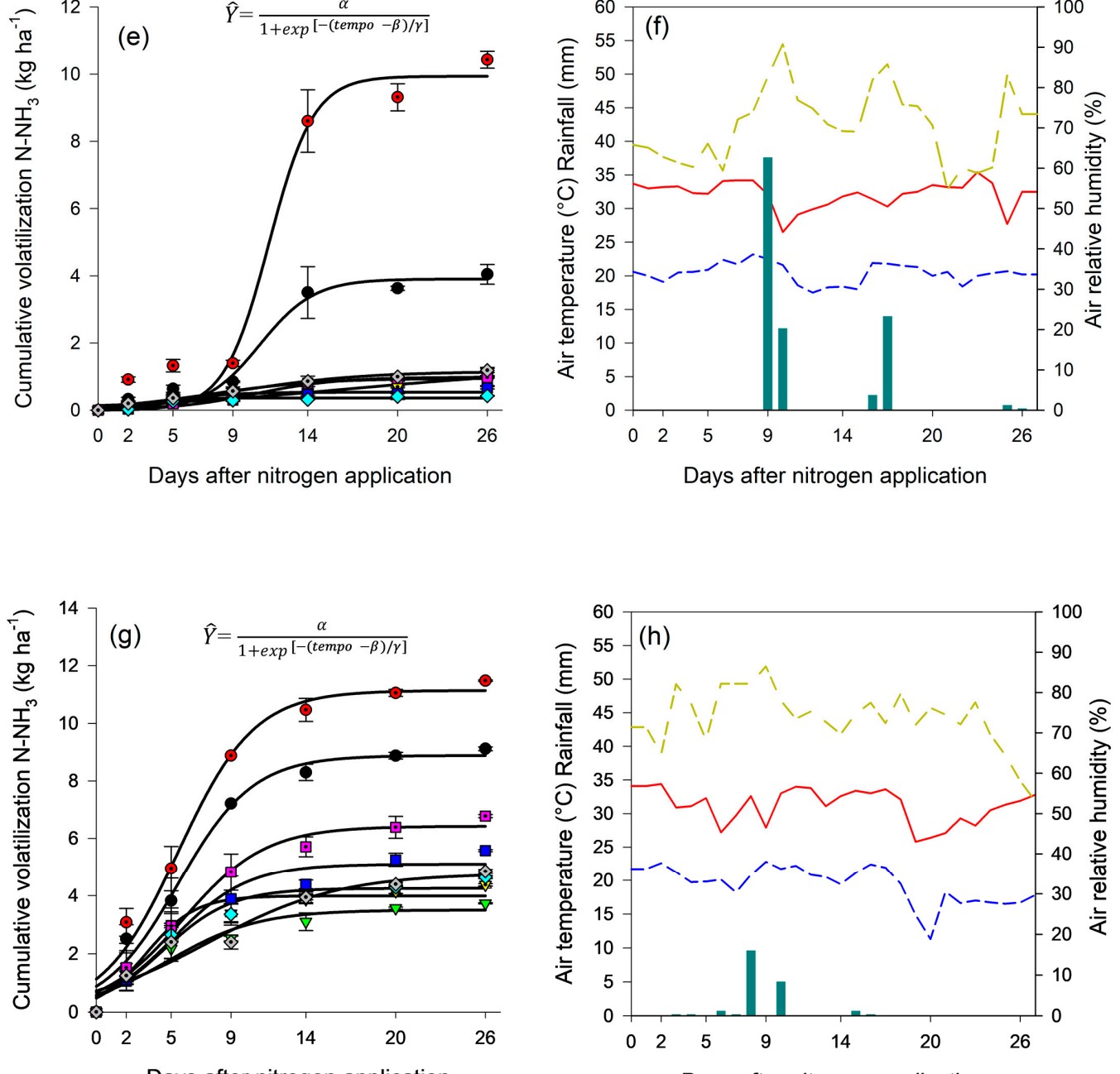

**Figure 3.** Ammonia volatilization (**a**,**c**,**e**,**g**) and climatic conditions (**b**,**d**,**f**,**h**) with application of 25 and 50 kg N ha$^{-1}$, referring to $\frac{1}{4}$ of the total rate of 100 kg ha$^{-1}$ and 200 kg N ha$^{-1}$, respectively. (**a**) 1st application (**c**) 2nd application (**e**) 3rd application (**g**) 4th application. Growing season 2018/2019.

The rainfall (9.7 mm) in the first days after fertilization in the fourth evaluation intensified ammonia losses from $Ur_{Conv100}$, to about 11.47 kg N ha$^{-1}$ (32% of applied N) (Figure 3g,h). The losses from $Ur_{Duromide100/200}$ were slightly less than from $Ur_{NBPT100/200}$ at both rates, with values very close to $AN_{100/200}$.

Figure 4 shows the values of ammonia volatilization after two forage cuts. In both evaluations and at both rates, ammonia volatilization from $Ur_{Conv}$ was higher. In turn, $Ur_{NBPT100/200}$ and $Ur_{Duromide100/200}$ proved to be efficient in reducing ammonia volatilization, reaching values similar to $AN_{100/200}$.

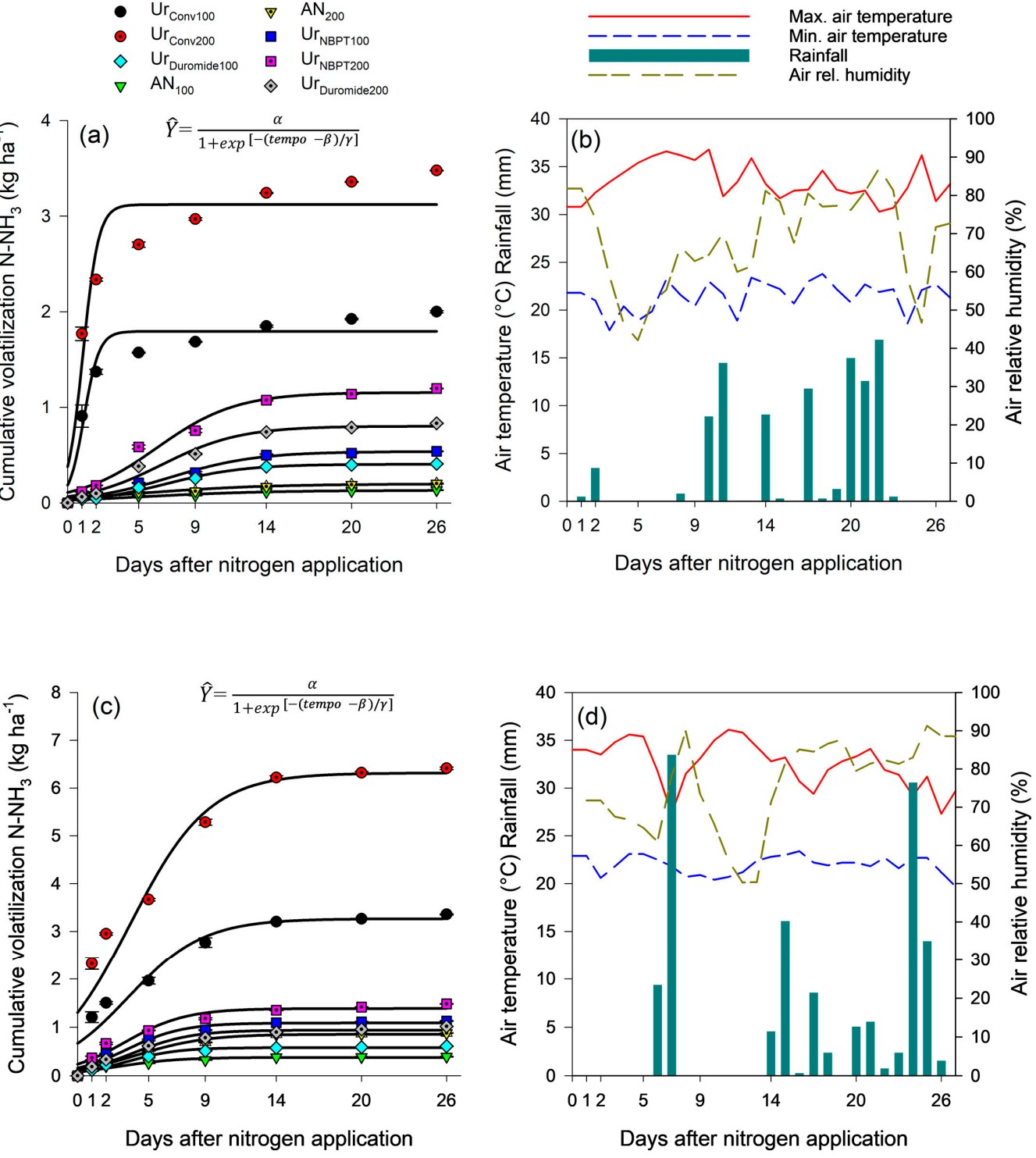

**Figure 4.** Ammonia volatilization (**a,c**) and climatic conditions (**b,d**) after application of 25 and 50 kg N ha$^{-1}$, corresponding to $\frac{1}{4}$ of the total rate of 100 and 200 kg N ha$^{-1}$, respectively. (**a**) 2nd application (**c**) 4th application. Growing season 2019/2020.

At the second fertilizer rate application, the first rainfall occurred one day after fertilization, but was insufficient to solubilize the fertilizers and incorporate them into the soil, so Ur$_{Conv100}$ reached 7.8% volatilization (2.00 kg N ha$^{-1}$) and Ur$_{Conv200}$ 6.8% (3.47 kg N ha$^{-1}$) (Figure 3a). Compared to conventional urea, the responses to AN in the second growing season (2019/2020) in the second cycle provided reductions of 92.7% and 93.9%, respectively, in response to N rates of 100 and 200 kg N ha$^{-1}$. For sources with urease inhibitors,

losses from $Ur_{Duromide100/200}$ were reduced by 77.6% and 74.3 and from $Ur_{NBPT\ 100/200}$ by 70.3% and 63.1%, in response to the same respective rates (Table 2).

**Table 1.** Parameters of the nonlinear (logistic) model fitted to the cumulative $NH_3$-N losses for N rates of 100 and 200 kg ha$^{-1}$, split into four applications, and reduction of $NH_3$-N losses in comparison with urea. Growing season 2018/2019.

| Treatments | Growing Season 2018/2019 Cycle | Parameters | | | | MDL | Reduction of NH$_3$-N Losses in Comparison with Urea (%) |
|---|---|---|---|---|---|---|---|
| | | $\alpha$ kg NH$_3$-N ha$^{-1}$ | $\Gamma$ | $\beta$ Day | R$^2$ | kg ha$^{-1}$ day$^{-1}$ NH$_3$-N | |
| Ur$_{Conv100}$ | 1 | 4.9 | 0.31 | 1.9 | 0.96 | 3.86 | - |
| | 2 | 4.26 | 0.1 | 1.77 | 0.97 | 10.65 | - |
| | 3 | 3.9 | 1.81 | 10.79 | 0.96 | 0.53 | - |
| | 4 | 8.88 | 2.49 | 5.48 | 0.97 | 0.89 | - |
| AN$_{100}$ | 1 | 0.27 | 3.1 | 8.83 | 0.99 | 0.02 | 94.4 |
| | 2 | 0.16 | 5.23 | 11.87 | 0.95 | 0.00 | 96.2 |
| | 3 | 0.95 | 3.69 | 7.39 | 0.93 | 0.06 | 75.6 |
| | 4 | 3.5 | 2.99 | 4.77 | 0.89 | 0.29 | 60.6 |
| Ur$_{NBPT100}$ | 1 | 3.18 | 1.48 | 5.38 | 0.97 | 0.53 | 35.1 |
| | 2 | 1.15 | 6.25 | 11.74 | 0.89 | 0.04 | 73.0 |
| | 3 | 0.53 | 2.01 | 4.76 | 0.85 | 0.06 | 86.4 |
| | 4 | 5.1 | 2.48 | 5.16 | 0.92 | 0.51 | 42.6 |
| Ur$_{Duromide100}$ | 1 | 2.33 | 1.32 | 5.02 | 0.96 | 0.44 | 52.4 |
| | 2 | 0.77 | 3.43 | 5.47 | 0.88 | 0.05 | 81.9 |
| | 3 | 0.36 | 0.93 | 4.13 | 0.89 | 0.09 | 90.8 |
| | 4 | 4.25 | 2.26 | 4.47 | 0.93 | 0.47 | 52.1 |
| Ur$_{Conv200}$ | 1 | 13.42 | 0.29 | 1.93 | 0.99 | 11.41 | - |
| | 2 | 8.17 | 0.09 | 1.77 | 0.99 | 22.69 | - |
| | 3 | 9.93 | 1.48 | 11.4 | 0.96 | 1.67 | - |
| | 4 | 11.13 | 2.52 | 5.5 | 0.97 | 1.10 | - |
| AN$_{200}$ | 1 | 0.31 | 2.99 | 9.62 | 0.99 | 0.02 | 97.7 |
| | 2 | 0.16 | 5.23 | 11.9 | 0.95 | 0.00 | 98.0 |
| | 3 | 1.28 | 8.08 | 16.92 | 0.89 | 0.03 | 87.1 |
| | 4 | 3.99 | 1.66 | 3.3 | 0.92 | 0.60 | 64.2 |
| Ur$_{NBPT200}$ | 1 | 6.14 | 1.54 | 5.72 | 0.98 | 0.99 | 54.2 |
| | 2 | 1.52 | 3.6 | 5.7 | 0.91 | 0.10 | 81.4 |
| | 3 | 0.98 | 2.8 | 10.5 | 0.98 | 0.08 | 90.1 |
| | 4 | 6.42 | 2.74 | 5.83 | 0.96 | 0.58 | 42.3 |
| Ur$_{Duromide200}$ | 1 | 3.39 | 1.14 | 4.7 | 0.96 | 0.74 | 74.7 |
| | 2 | 1.06 | 3.3 | 5.71 | 0.84 | 0.08 | 87.0 |
| | 3 | 1.15 | 4.28 | 9.26 | 0.96 | 0.06 | 88.4 |
| | 4 | 4.77 | 4.37 | 7.47 | 0.9 | 0.27 | 42.9 |

MDL: maximum daily $NH_3$-N loss.

After the fourth fertilization, the first rainfall occurred only five days after fertilization, showing that during this rainless period, volatilization was very low. At the onset of rains, volatilization became intense in the treatments with $Ur_{Conv200}$, reaching losses of approximately 13% in 14 days after fertilization, while losses from $Ur_{Duromide100/200}$, $Ur_{NBPT100/200}$ and $AN_{100/200}$ were less throughout the entire period. In this experiment, at rates of 25 and 50 kg N ha$^{-1}$ as AN, volatilization reduction was smaller than in the previous cycle, with 88.6% and 86.5%, respectively, while volatilization from $Ur_{NBPT100/200}$ was reduced by 57.6% and 82.9% and from $Ur_{Duromide100/200}$ by 82.5% and 85.1%, respectively (Table 2). In summary, of the total of twelve volatilization comparisons, in eleven of them $Ur_{Duromide}$ resulted in less losses than $Ur_{NBPT}$.

**Table 2.** Parameters of the nonlinear (logistic) model fitted to the cumulative $NH_3$-N losses for N rates of 100 and 200 kg ha$^{-1}$, split into four applications, and reduction of $NH_3$-N losses in relation to urea. Growing season 2019/2020.

| Treatments | Growing Season 2019/2020 Cycle | Parameters | | | | MDL | Reduction of $NH_3$-N Losses in Comparison with Urea (%) |
|---|---|---|---|---|---|---|---|
| | | $\alpha$ kg $NH_3$-N ha$^{-1}$ | $\gamma$ | $\beta$ Day | $R^2$ | kg ha$^{-1}$ day$^{-1}$ $NH_3$-N | |
| Ur$_{Conv100}$ | 2 | 1.79 | 0.53 | 1.15 | 0.93 | 0.23 | - |
| | 4 | 3.26 | 2.65 | 3.6 | 0.9 | 2.15 | - |
| An$_{100}$ | 2 | 0.13 | 4.37 | 5.34 | 0.83 | 0.14 | 92.74 |
| | 4 | 0.37 | 2.15 | 2.83 | 0.88 | 0.19 | 88.65 |
| Ur$_{NBPT100}$ | 2 | 0.53 | 3.03 | 7.23 | 0.98 | 0.40 | 70.39 |
| | 4 | 1.38 | 2.13 | 3.34 | 0.91 | 0.73 | 57.67 |
| Ur$_{Duromide100}$ | 2 | 0.4 | 2.74 | 6.99 | 0.98 | 0.27 | 77.65 |
| | 4 | 0.57 | 0.01 | 1.53 | 0.96 | 0.00 | 82.52 |
| Ur$_{Conv200}$ | 2 | 3.12 | 0.54 | 1.07 | 0.89 | 0.42 | - |
| | 4 | 6.31 | 2.76 | 3.71 | 0.89 | 4.35 | - |
| An$_{200}$ | 2 | 0.19 | 4.56 | 4.86 | 0.79 | 0.21 | 93.91 |
| | 4 | 0.85 | 2.53 | 4.2 | 0.94 | 0.53 | 86.53 |
| Ur$_{NBPT200}$ | 2 | 1.15 | 2.79 | 6.29 | 0.96 | 0.80 | 63.14 |
| | 4 | 1.08 | 1.96 | 3.5 | 0.95 | 0.52 | 82.88 |
| Ur$_{Duromide200}$ | 2 | 0.8 | 2.75 | 6.6 | 0.96 | 0.55 | 74.36 |
| | 4 | 0.94 | 2.1 | 3.92 | 0.95 | 0.48 | 85.10 |

MDL: maximum daily $NH_3$-N loss.

### 3.2. Nitrogen Use Efficiency for Dry Matter Production

Nitrogen use efficiency in Marandu dry matter production in the first growing season ($p < 0.001$) was influenced by N sources and rates. The N use efficiency of Ur$_{Duromide200}$ (50.6 kg dry matter/kg N) was greater than that of Ur$_{Conv100}$ (5.1 kg dry matter/kg N). In turn, Ur$_{Duromide}$ 100 (37.1 kg dry matter/kg N) and Ur$_{NBPT100/200}$ (30 and 34.8 kg dry matter/kg N, respectively) also proved more efficient than Ur$_{Conv100/200}$ (kg dry matter/kg N). The efficiency of dry matter production in response to fertilization with AN$_{100/200}$ was close to that observed for Ur$_{Conv100/200}$ (Figure 5).

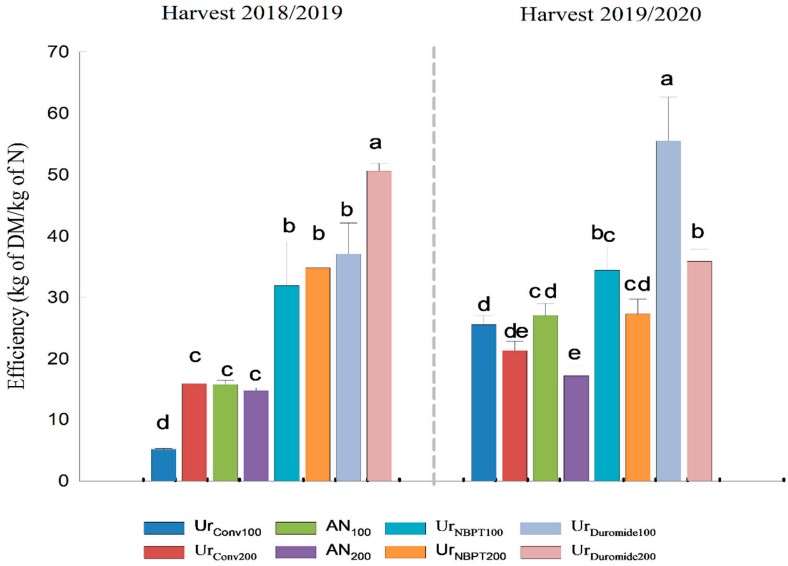

**Figure 5.** Dry matter production efficiency of *Urochloa brizantha* cv. Marandu fertilized with nitrogen rates and sources for pasture formation and maintenance. Growing seasons 2018/2019 and 2019/2020. Means followed by distinct vertical letters were significantly different according to Tukey's test ($p < 0.05$).

In the second year of evaluations, the forage used more N from $Ur_{Duromide100/200}$ than from $Ur_{Conv100/200}$ and $AN_{100/200}$ ($p < 0.001$). The efficiency of $Ur_{Duromide\ 100}$ was higher (55.5 kg dry matter/kg N) than that of the sources without urease inhibitor. In the case of $AN_{200}$, the production efficiency ratio was less (17.3 5 kg dry matter/kg N) (Figure 5).

The total dry matter and leaf production in response to N rates differed significantly between treatments, and the rate of 200 kg N ha$^{-1}$ differed for all sources, in both years of evaluation. In the first year, $Ur_{Duromide200}$ stood out from the other sources, with increases of 73.6% dry matter production and 20.4% leaf production in relation to $Ur_{Conv100/200}$. On the other hand, compared to $Ur_{Conv100/200}$, $Ur_{Duromide\ 100}$ provided an increase of 47.6% dry matter and 33% leaf production (Figure 6a,c).

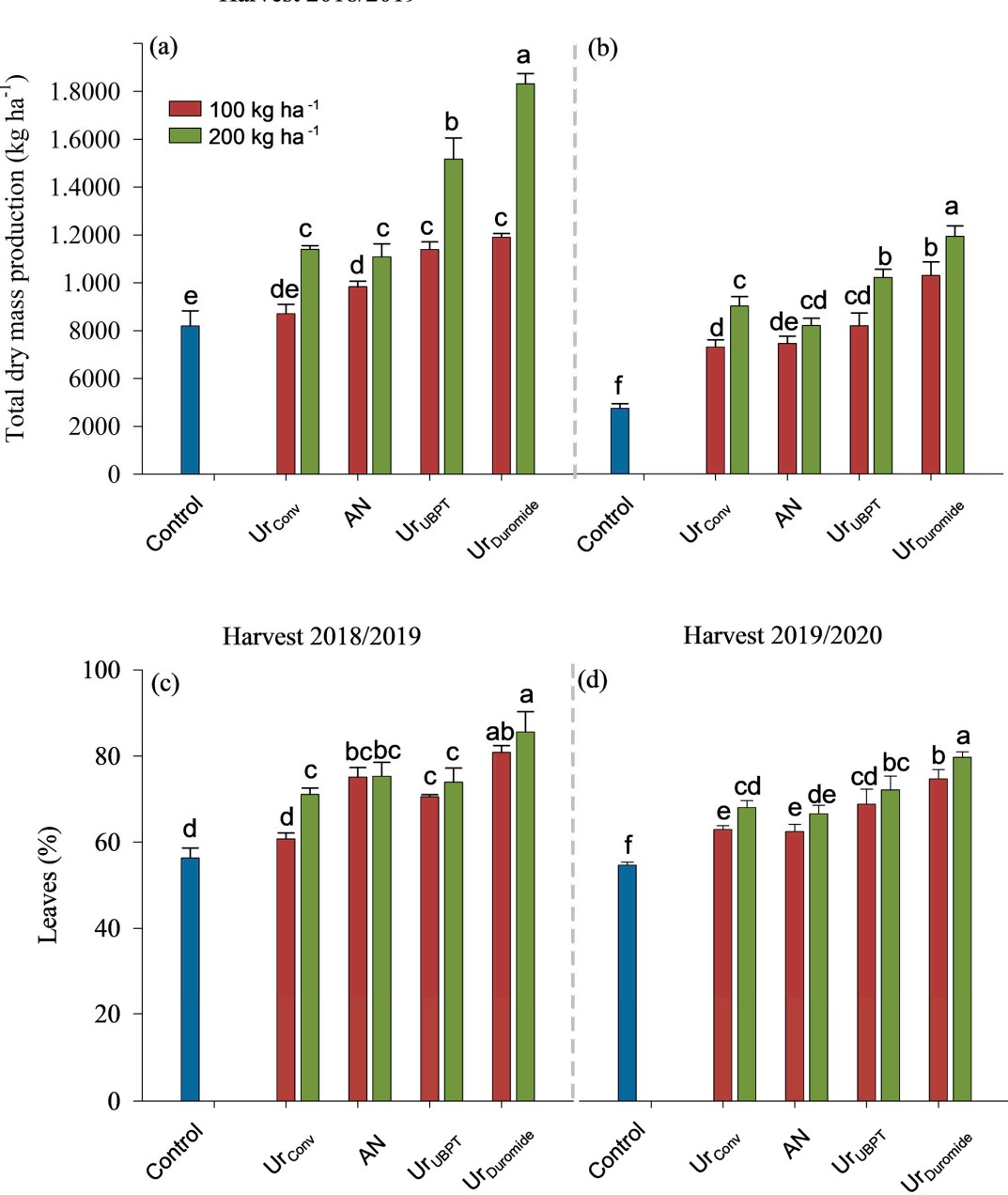

**Figure 6.** Total dry matter production and relative leaf production of *Urochloa brizantha* cv. Marandu, in two growing seasons, fertilized with nitrogen sources and rates. Means followed by distinct vertical letters were significantly different according to Tukey's test ($p < 0.05$).

In the second year, dry matter and leaf production were relatively less than in the previous year (2018/2019), although dry matter production in response to $Ur_{Duromide100/200}$ still exceeded that of $Ur_{Conv100/200}$, with about 56.3% dry matter, i.e., more than in the previous year at 100 kg N ha$^{-1}$. The increase at 200 kg N ha$^{-1}$ was 41.3% in comparison with $Ur_{Conv}$ (Figure 6b,d)

In the second year, nitrogen fertilization induced an increase in leaf percentage over the control, mainly in response to $Ur_{Duromide200}$, resulting in an increase of 17.1% in leaf production (Figure 6d).

*3.3. Correlation*

Pearson's correlation analysis between pasture production parameters and nitrogen rates detected a positive correlation between dry matter production and leaf production, leaf dry matter, reduction of N losses and N use efficiency. Total dry matter and N loss percentage were negatively correlated, i.e., dry matter increases as N losses decrease. This result was also observed for leaf percentage and N use efficiency, which increased with decreasing N losses (Table 3).

**Table 3.** Pearson's correlation of total dry matter production and leaf production with parameters of N dynamics in a *Urochloa brizantha* cv. Marandu pasture, in two growing seasons and in response to two annual N rates, applied in four split rates.

| Parameters | 2018/2019 | | 2019/2020 | |
|---|---|---|---|---|
| | 100 (kg ha$^{-1}$) | 200 (kg ha$^{-1}$) | 100 (kg ha$^{-1}$) | 200 (kg ha$^{-1}$) |
| Total dry matter/% leaves | 0.87 | 0.88 | 0.98 | 0.99 |
| Total dry matter/leaf dry matter | 0.97 | 0.99 | 0.99 | 0.99 |
| Total dry matter/N loss reduction | 0.73 | 0.38 | 0.43 | 0.24 |
| Total dry matter/N use efficiency | 1.00 | 1.00 | 0.97 | 0.99 |
| Total dry matter/% N loss | −0.75 | −0.35 | −0.44 | −0.26 |
| % leaves/N loss reduction | 0.85 | 0.54 | 0.40 | 0.31 |
| % leaves/N use efficiency | 0.77 | 0.83 | 0.97 | 0.99 |
| % leaves/% N loss | −0.86 | −0.52 | −0.40 | −0.33 |
| N use efficiency/% N loss | −0.73 | −0.35 | −0.44 | −0.26 |

**4. Discussion**

In general, ammonia losses were high in the $Ur_{Conv}$ treatment (Figures 3 and 4), which contributed to reduce N availability, reflecting in lower dry matter production than of $Ur_{Duromide100/200}$ and $Ur_{NBPT100/200}$ (Figure 6). The high ammonia losses from $Ur_{Conv}$ can be explained by the greater saturation of the sites of urease enzyme action, due to the higher ammonium availability in soil fertilized with untreated urea, as stated by [29].

In both years, NH$_3$-N losses occurred as of the 2nd day after fertilizer application to the soil, except for the third cycle, with the onset of volatilization on the 10th day (Figure 3e,f). The amplitude of ammonia losses is very variable after urea application to the soil surface, and depends on the rates, sources and prevailing climatic conditions during the evaluation period. The hydrolysis rate of $Ur_{Conv}$ by the enzyme urease was higher in the first 2–3 days after fertilization, according to soil temperature, moisture and fertilizer volume applied per area [29].

The intense volatilization from the third fertilization plot of the first year may also have been caused by the collector, where $Ur_{Conv100/200}$ was possibly prevented from being incorporated in the area of volatilization measurement (inside the collector) by rainwater. The rain that fell around the collector probably caused surface wetting of the soil, favoring urea solubilization, but not enough for soil incorporation. With regard to the sources with urease inhibitor and ammonium nitrate, the values were much lower during the whole period, demonstrating the inhibition efficiency under adverse conditions (Figure 3e,f). According to Afshar et al. [30], the variation in the amount of volatilized NH$_3$ was significant, which may be a result of the measurement method and environmental conditions.

Indeed, the total emissions were possibly not fully captured because measurements were not made daily.

In all evaluations, losses ceased after the volatilization peak, which can be explained by the rainfall volume, which was insufficient for fertilizer aggregation in the soil, resulting in a stabilization of the losses after a period with higher soil water concentrations. This may have occurred due to the rapid hydrolysis of urea by urease, since losses were greater in the first seven days immediately after soil fertilization [30]. Similar results were reported for coffee, where the authors observed that up to 35% N can be lost from $Ur_{Conv}$ [31].

The results of this study confirmed that the amount of volatilized N was lower from sources with urease inhibitors and ammonium nitrate. This finding is in agreement with Otto et al. [19], who observed that fertilization with ammonium nitrate resulted in N volatilization losses of less than 1%. However, under favorable climatic conditions, the N fertilizers commonly used in agriculture, as is the case with $Ur_{Conv}$ and AN, can be used as fertilizers for pasture growth. However, fertilizers containing ammonia or urea promote soil acidification in the production system, especially when used at high rates [32].

Losses from $Ur_{Duromide}$ were a little lower than from $Ur_{NBPT}$ at both annual rates and very close to AN, and also promoted a slightly delayed onset of N losses. A delay in the initial loss of N from $Ur_{NBPT}$ and $Ur_{Duromide}$ was confirmed by Cassim et al. [33], increasing the chances that urea is incorporated into the soil by rain, for example, which would reduce $NH_3$ volatilization. However, it is worth remembering that the ammonia loss measurement method in this study was not able to detect losses by denitrification and leaching, which occur more easily from AN.

The reduction in ammonia loss from fertilizer with $Ur_{NBPT}$ was greater than that found by Souza [34]. Comparing the peak of fertilizer volatilization, the $NH_3$ loss from $Ur_{NBPT}$ was reduced by 77% in relation to $Ur_{Conv}$ (fourth day after fertilizer application). Other studies also concluded that the urease inhibitor reduces $NH_3$ losses by 54% in comparison with $Ur_{Conv}$ [29]. Sources with urease-inhibiting technologies ($Ur_{Duromide}$ and $Ur_{NBPT}$) were more efficient than $Ur_{Conv}$, with reductions of over 50% compared to $Ur_{Conv}$. According to Cassim et al. [33], $Ur_{Duromide}$ and $Ur_{NBPT}$ were extremely efficient in reducing $NH_3$ volatilization (reductions of 35–54%) at 45 and 90 kg N ha$^{-1}$. In comparison with $Ur_{Conv}$, the new Duromide stabilizing technology reduced $NH_3$ losses by up to 33%, compared to NBPT alone.

The dry matter production efficiency of *Urochloa brizantha* cv. Marandu, in the first growing season with $Ur_{Duromide200}$ was more efficient than that of $Ur_{Conv100}$, however in the second year, the efficiency of $Ur_{Duromide200}$ as well as $AN_{200}$ decreased with increasing nitrogen rates. According to Rowlings et al. [35], a reduction in fertilization efficiency is common with increasing N rates. Nitrogen rates and sources positively influence dry matter and leaf production of Marandu grass, and should be taken into account to achieve improved forage quantity and quality [36,37]. A study on N fertilization of Mombasa grass also showed an increase in dry matter accumulation, in addition to increasing nutrient accumulation and improving pasture maintenance [38]. It is worth emphasizing that not only volatilization was reduced by the new $Ur_{Duromide}$ technology, but that leaf percentage, dry matter production and N use efficiency were also increased.

## 5. Conclusions

The urease inhibitors $Ur_{NBPT}$ and $Ur_{Duromide}$ reduced N losses by ammonia volatilization, to values similar to those observed for ammonium nitrate. Ammonia losses from $Ur_{Duromide}$ tend to be lower than from $Ur_{NBPT}$. In the separate evaluation of each cut, in the 2018/2019 and 2019/2020 growing seasons, volatilization losses from $Ur_{Conv100/200}$ were greater than from AN, $Ur_{NBPT}$ and $Ur_{Duromide}$.

The total dry matter and the leaves dry matter production of Marandu grass in the first and second year, were highest in response to $Ur_{Duromide100/200}$ in comparison with the other sources ($Ur_{Conv}$, AN and $Ur_{NBPT}$).

It is concluded that compared to conventional urea, $Ur_{NBPT}$ and $Ur_{Duromide}$ reduce N losses by $NH_3$ volatilization. The total dry matter and leaf yield of Marandu grass in the first and second year were higher in response to $Ur_{Duromide100/200}$ compared to the other sources ($Ur_{Conv}$, AN and $Ur_{NBPT}$).

Thus, the combination of duromide and NBPT is a very promising technology for reducing losses by ammonia volatilization and greater utilization of nitrogen from the fertilizer by the crop and higher productivity.

**Author Contributions:** Conceptualization, J.B.C. and R.H.; methodology J.B.C., C.L.B.d.O., A.d.S.B., N.d.L.D., G.C.M., I.V.R. and R.H.; software and validation, J.B.C., R.H. and C.L.B.d.O.; Resource, R.H.; formal analysis, J.B.C., C.L.B.d.O., A.d.S.B., N.d.L.D., G.C.M. and R.H.; investigation, J.B.C., C.L.B.d.O.; data curation, J.B.C., C.L.B.d.O.; writing—original draft preparation, J.B.C. and C.L.B.d.O.; writing—review and editing, J.B.C., C.L.B.d.O., A.d.S.B., J.F.d.S. and R.H.; visualization, J.B.C. and R.H.; supervision, R.H. All authors have read and agreed to the published version of the manuscript.

**Funding:** This research received no external funding.

**Data Availability Statement:** The data presented in this study are available on request from the corresponding author.

**Conflicts of Interest:** The authors declare no conflict of interest.

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
