# Peer review of "Ammonia Volatilization and Marandu Grass Production in Response to Enhanced-Efficiency Nitrogen Fertilizers"

_agronomy, doi:10.3390/agronomy13030837_

Round 1
Reviewer 1 Report
Methodology
The description of the measurement of NH3 can be improved. The explanation is generally clear but the citation to the methodology was published in a local journal in Portuguese, so not accessible to most readers. It would therefore be useful to have a little more detail of the method in this paper, e.g. a very brief summary of any recovery tests carried out and/or comparison with internationally recognised NH3 measurement techniques, e.g. micrometeorological mass balance.
Most of the conclusions are supported by the results, but I do not agree that 'The urease inhibitors UrNBPT and UrDuromide reduced N losses by ammonia volatilization, to values similar to those observed for ammonium nitrate.' in every case. Figures 3a and 3c indicate losses from AN were little more than zero while c. 2 kg/ha were emitted despite the addition of inhibitors. Certainly the inhibitors substantially reduced NH3 emission and Duromide was more effective than NBPT. The authors should leave their conclusions to the inhibitors producing substantial reductions and in some cases (Fig 4c) little different to AN
References
I'm not convinced reference 20 is relevant to estimating cumulative NH3 emissions (lines 150, 291)
Reference number 7 in the reference list is not cited in the paper, the reference numbers go from 6 to 8.
The presentation of results can be improved a little but generally fine.
Some English changes required and these are listed below.
Other points
Line 38, replace 'have shown satisfactory results' with 'increase urea efficiency'.
Lines 47-48. I suggest a reference be cited to justify this statement.
Line 103, not sure what is meant by 'exchanged twice'. Do you mean twice per day? Please clarify.
Figure 2, the line indicating 'Foam' extends to the rigid wire, hence the line needs to more clearly indicate the foam.
Line 125, explain what you mean by 'best' treatment. Do you mean when one of the treatments reached 28 cm?
Lines 128-9, can sampling be both random and representative? Please explain.
Lines 190-191, it should be 'efficiency of urease inhibitor with urea'.
Minor
Line 13, replace 'installed' with 'took place' and delete ', evaluated'.
Line 15, replace 'Ureia' with 'Urea'.
Line 17 and elsewhere, replace 'highest' with 'greatest' since you are reporting amounts not elevation.
Line 18 and elsewhere, replace 'lower' with 'less'.
Line 31, replace 'highly' with 'very'.
Line 32, replace 'at high and frequent rates' with 'frequently with large amounts in total'.
Line 75, replace 'replications' with 'replicates'.
Line 83, delete 'of evaluations'.
Line 105, replace 'fit' with 'fitted'.
Line 120, subscript numbers needed for 'H2SO4'.
Line 124, replace 'performed' with 'taken'.
Line 166, replace 'evaluation cycle' with 'experiment' and following lines use 'experiment'.
Line 173 and elsewhere, there is no need to cite the % reductions to two decimal places. I suggest you be consistent with the way the results are presented in Table 1.
Line 181, replace 'in response' with 'compared'.
Line 189, and elsewhere, replace 'lowest' with 'least'.
Line 193, and elsewhere, replace 'lower' with 'less'.
Lines 400, 402, 404, 426, 443, 444. Some words all in capitals and not consistent with other entries.
Author Response
Dear Reviewer
Thank you for your time and your comments on our article.
I am responding point by point to what you suggested.
For better understanding I have attached a file

Reviewer 2 Report
This manuscript is interesting and study about ammonia volatilization and marandu grass production in response to enhanced-efficiency nitrogen fertilizers. However, there are a number of issues in current form which requires revision. Concept of the manuscript is up to the mark but the development is poor according to me. Information related to results needs to be compressed and some results can move to seperate supplimentary material file for improve readability. Plenty of flaws are there in methods and results and are not well described as well. Conclusion need to streamline and rewite for improve readability only focus on real findings of the study and its implications.
My specific comments are attached in zip file

Author Response

(The authors gave the same response as above.)

Reviewer 3 Report
This manuscript describes ammonia volatilization after fertilization with urea, coated urea, and ammonium nitrate in mandu grass. I found a few English errors within the text that will need to be fixed. I liked the graphs that showed volatilization over time for each fertilization along with climate conditions. I think the discussion could be expanded to compare more related research. Can you make a suggestion for the type of fertilizer and rate recommended due to this research? I believe that this manuscript should be accepted after minor revisions.
Line 21: What is DM?
Lines 32-33: Rephrase this sentence because it does not make sense as written.
Line 37: Change the period in the middle of the sentence to a comma.
Line 45: Remove either “caused” or “driven”
Lines 67-68: Can you describe this soil type for those not familiar with this classification system?
Author Response
Dear Reviewer
Thank you for your time and comments on our article.
I'm responding point by point to what you suggested.
